# Epidemiology of substance use and mental health disorders among forced migrants displaced from the MENAT region: A systematic review and meta-analysis protocol

**Maryam Kazemitabar** [1,2]*, **Kate Nyhan**[3,4], **Natalie Makableh**[5], **Rebecca Minahan-Rowley**[1], **Malak Ali**[6], **Mayyada Wazaify** [6], **Jeanette Tetrault**[1], **Kaveh Khoshnood**[5]

**1** Department of Internal Medicine, Yale School of Medicine, New Haven, CT, United States of America, **2** VA Connecticut Healthcare System, West Haven, CT, United States of America, **3** Harvey Cushing / John Hay Whitney Medical Library, Yale University, New Haven, CT, United States of America, **4** Department of Environmental Health Sciences, Yale School of Public Health, Yale University, New Haven, CT, United States of America, **5** Department of Epidemiology of Microbial Diseases, Yale School of Public Health, New Haven, CT, United States of America, **6** Department of Biopharmaceutics and Clinical Pharmacy, Faculty of Pharmacy, The University of Jordan, Amman, Jordan

* maryam.kazemitabar@yale.edu

**Data Availability Statement:** No datasets were generated or analyzed during the current study. All

## Abstract

### Background

Understanding the epidemiology of substance use disorders (SUDs) and mental health disorders (MHDs) among forced migrants from the Middle East and North Africa and Türkiye (MENAT) region is of utmost importance given their distinct challenges and vulnerabilities. Existing research on the topic is substantial; however, comprehensive systematic reviews are limited. We aim to conduct a systematic literature review to bridge this gap, providing a thorough understanding of SUDs and MHDs epidemiology among this population.

### Methods

The systematic review will include original, peer-reviewed research articles published in English or Arabic between 2013 and 2023. It will encompass qualitative, quantitative, and mixed-methods studies focusing on SUDs and/or MHDs among forced migrants from the MENAT region. A Medline (via Ovid) search strategy was developed and will be translated into databases of EMBASE (via Ovid), Web of Science Core Collection (as licensed at Yale), and CINAHL. Risk of Bias in the included studies will be assessed using the Joanna Briggs Institute (JBI) checklist for observational studies and the Cochrane Risk of Bias (ROB) 2 tool for intervention studies. Meta-analysis using random-effects models will be conducted and subgroup analyses based on studies' data will be performed. The systematic review will be conducted based on the framework introduced by the Joanna Briggs Institute. Additionally, the PRISMA-P extension for systematic reviews was used to report the present protocol. The present systematic review protocol has been registered with PROSPERO, an

relevant data from this study will be made available upon study completion.

**Funding:** The author(s) received no specific funding for this work.

**Competing interests:** The authors have declared that no competing interests exist.

international prospective register of systematic reviews. The review's registration ID is CRD42021286882.

## Discussion

The systematic review aims to explore, identify and synthesize the evidence to reveal the epidemiology of SU and/or MH of forced migrants of the MENAT region as described in this protocol. In sum, the review will provide evidence to fill the gap in the literature and help the stakeholders, policymakers, and health providers to expand current knowledge of the prevalence and risk factors of SU and/or MH in humanitarian settings.

## Introduction

Substance use disorders (SUDs) and mental health disorders (MHDs) are among the leading causes of fatal ailments [1]. A report by "Our World in Data" estimates that about 11.8 million people lose their lives annually due to substance use and related disorders worldwide [2]. Aside from health issues, SUDs, and MHDs can lead to various problems such as homelessness, criminal legal involvements, incarceration, transmission of HIV due to injecting drugs, high-risk sexual behaviors, unemployment, and dependence on welfare [3, 4]. Additionally, SUDs and MHDs limit an individual's average performance in activities and might even lead to suicide and death [5]. The exorbitant costs associated with SUDs and MHDs is an economic burden for governments [6, 7].

According to the United Nations Human Rights Council, over 84 million people were forcibly displaced in mid-2021 due to persecution, war, or violence [8, 9]. Many of whom are likely struggling with MHDs and/or SUDs. The Middle East and North Africa and Türkiye (MENAT) region has experienced lots of unrest in the last several decades, resulting in forced migration for numerous people. Forced migrants in this region are forcibly displaced due to persecution, war, violence, and natural disasters such as floods and storms [10]. The Arab Spring protest and the subsequent state crackdowns, coups, and wars that followed the initial protests played a significant role in the displacement, such that caused more than 3.5 million people to get internally displaced; ever since, the numbers have increased to more than three times. Moreover, the MENAT region had approximately more than 7.8 million internally displaced people seeking asylum by 2021 [11].

Forced migrants have potentially high risks of SUDs and MHDs since they are exposed to challenging conditions that could have a lifelong impact. Forced migration, availability of illicit drugs in refugee camps, exposure to conflict, exploitation and mental, physical, and sexual abuse, trafficking, difficult living conditions, loss of resources, division of family, separation anxiety, isolation, stress of relocation, challenges associated with learning a new language, and unemployment are among common challenges that forced migrants encounter. Such events can trigger mental health disorders such as posttraumatic stress disorder (PTSD), anxiety, and depression [12, 13]. In addition, struggling with consequences and stress from forced displacement can lead individuals to use substances. Studies show that SUDs and MHDs frequently occur among forced migrants, such that about 50% of the individuals who experience a mental health disorder during their life will also experience a substance use disorder and vice versa [14, 15]. This population uses substances to cope with acculturation challenges, social and economic inequalities as well as traumatic experiences due to loss of livelihood, torture, and family separations [16].

Studying the epidemiology of SUDs and MHDs among forced migrants from the MENAT region is crucial due to their unique challenges and vulnerabilities. There is a substantial body of research on SUDs [16, 17] and especially MHDs [18–22] from the MENAT region. However, the availability of systematic reviews that comprehensively collect and analyze this data is limited. Conducting a systematic literature review will help bridge this gap by providing a comprehensive synthesis of the existing research, allowing for a more thorough understanding of the epidemiology of SUDs and MHDs among this population. In this systematic review protocol, we aim to fill this gap in the literature and help the stakeholders, policymakers, and health providers to expand their knowledge of the prevalence and risk factors of substance use and mental health disorders in humanitarian settings. As a result, we aim to find the answers to the following questions:

## Primary review questions

1. What is the prevalence of SUDs and/or MHDs among forced migrants displaced from the MENAT region?
   Expected outcome: The proportion or rate of individuals among forced migrants displaced from the MENAT region who have been diagnosed with SUDs and/or MHDs. Examples of specific disorders include but are not limited to depression, anxiety, PTSD, substance dependence, or substance abuse.

2. What are the causes and risk factors of SUDs and/or MHDs among forced migrants displaced from the MENAT region?
   Expected outcome: Factors, circumstances, or events contributing to the development or increased risk of SUDs and/or MHDs among forced migrants displaced from the MENAT region. This may include factors such as exposure to conflict, trauma, displacement, loss of resources, social isolation, or environmental stressors.

3. What prevention and intervention considerations are provided to forced migrants with SUDs and/or MHDs displaced from the MENAT region?
   Expected outcome: Strategies, approaches, or interventions to prevent or address SUDs and/or MHDs among forced migrants displaced from the MENAT region. This includes preventive measures, mental health support, treatment options, psychosocial interventions, access to healthcare services, or policy considerations.

## Secondary review questions

1. What types of substances do forced migrants with SUDs and/or MHDs displaced from the MENAT region use?
   Expected outcome: The specific substances, such as alcohol, opioids, stimulants, or other drugs, used by forced migrants with SUDs and/or MHDs who have been displaced from the MENAT region.

2. What types of mental disorders are prevalent among forced migrants with SUDs and/or MHDs displaced from the MENAT region?
   Expected outcome: The specific mental disorders, such as depression, anxiety disorders, PTSD, or other psychiatric conditions, among forced migrants with SUDs and/or MHDs who have been displaced from the MENAT region.

## Methods

### Design of the systematic review

This systematic review protocol has been registered with PROSPERO, a global database for prospective registration of systematic reviews. The review has been assigned the registration ID of CRD42021286882. The current study uses the framework introduced by the Joanna Briggs Institute (JBI). The JBI framework is a comprehensive approach to conducting systematic reviews of evidence. It includes nine stages to the review process [23]:

1. Defining the review question: This involves developing a well-defined research question or objective.

2. Developing inclusion and exclusion criteria: This involves identifying the specific criteria that will be used to determine which studies will be included in the review.

3. Search strategy: Develop a comprehensive search strategy to identify all relevant studies for the review.

4. Study selection: Screen the search results against the inclusion and exclusion criteria and select the studies that meet these criteria.

5. Quality appraisal: Assess the quality of the included studies to determine their validity and reliability.

6. Data extraction: Extract relevant data from the included studies.

7. Data synthesis: Analyze and synthesize the data from the included studies to identify patterns and draw conclusions.

8. Assessing certainty of evidence: Evaluate the overall certainty of the evidence and the strength of the conclusions that can be drawn from it.

9. Reporting: Report the results of the review clearly and transparently.

Besides the JBI framework, the 'Preferred Reporting Items for Systematic Reviews and Meta-Analysis Protocols (PRISMA-P; [24]) extension for systematic reviews was used to report the present protocol.

### Eligibility criteria

**Participants.**  The participants of the studies include forced migrants with SUDs and/or MHDs from one of the MENAT countries without any age or gender limitation. Forced migration involves refugees and displaced populations compelled to leave due to war, conflict, persecution, disasters, development-induced displacement, smuggling, human trafficking, and environmental factors [25]. We will include forced migrants such as refugees, asylum seekers, and internally displaced persons in our review.

The systematic review will include all qualitative, quantitative studies, and mixed-methods studies focusing on SUDs and/or MHDs among forced migrants displaced from the MENAT region. The quantitative prevalence estimates of SUDs and MHDs should be classified by the latest version of Diagnostic and Statistical Manual of Mental Disorders (DSM) [26] which is DSM-5-TR. Included studies must be original and peer-reviewed research articles. Studies must provide original data; therefore, commentaries, editorials, letters, and opinions are excluded. The studies' language must be English or Arabic. We will include studies published from 2013 to 2023 (the last ten years). We use the United Nations Human Resources Office of the High Commissioner (HROHC) classification for included countries in the MENA region

[27] plus Türkiye (MENAT). Countries include Algeria, Bahrain, Egypt, Iran, Iraq, Israel, Jordan, Kuwait, Lebanon, Libya, Morocco, Palestine, Oman, Qatar, Saudi Arabia, Syria, Tunisia, Türkiye, United Arab Emirates, and Yemen. Studies focused on people from the MENAT region who have been forcibly displaced by conflict, persecution, and/or natural disaster, including international displacement as well as internal displacement will be included.

**Information sources.** Database selection and search will be designed collaboratively by domain experts and a medical librarian. Electronic databases of MEDLINE, EMBASE (both via Ovid), Web of Science Core Collection, CINAHL COMPLETE, Global Index Medicus, and The Index Medicus for the Eastern Mediterranean Region (IMEMR) will be searched. A hand search through WHO regional database and Google Scholar will also be conducted. Additionally, research articles published in journals not covered by the databases we searched can still be located using citation chaining in this study. We will perform this process using the Citation Chaser tool [28] and the bibliographic database Lens. This database incorporates publication details from the OpenAlex database/dataset, known for its strong coverage of open-access journals in low- and middle-income countries [29]. Citation Chaser is an automated tool that streamlines the process of "citation chasing" in systematic reviews. It uses the Lens.org API to quickly retrieve lists of references from various studies and identify articles that cite a specific study. This eliminates the manual effort traditionally required for cross-referencing and enhances accuracy. The tool can generate lists of both referenced and citing records from sources like PubMed, PubMed Central, CrossRef, Microsoft Academic Graph, and CORE, making systematic review searches more efficient. The steps involved in conducting citation chaining are:

1. Identifying a relevant article or resource.

2. Looking at the references cited in the article to identify other potentially relevant sources.

3. Examining the reference list of each identified article to identify additional relevant sources.

4. Repeating this process until no new relevant sources are found.

5. Evaluating the relevance and qualifying of each identified source.

6. Incorporating the new sources into our research and citing them appropriately.

## Search strategy

The research team members developed the search terms through a comprehensive list of search terms focused on substance use, mental health, forced migrants and MENAT region countries' relevant terms and keywords. We conducted a preliminary search to find the appropriate terms and keywords. Table 1 represents the search strategy developed for the Medline Ovid database. The search strategy was peer-reviewed by an independent medical librarian.

## Screening process

A public health librarian (KN) was responsible for designing and testing the search queries and an independent medical librarian (TM) peer-reviewed the search strategies. Those results will then be exported from databases and imported to Covidence and duplicates will be removed. The titles and abstracts returned will be screened. Records will be screened for title and abstract by independent reviewers and then, the reviewers will read and screen the full texts of the selected studies to determine whether they meet the eligibility criteria for final inclusion (MK, NM, RM-R, MA). Screening will be done in duplicate. Conflicts will be

**Table 1. Search strategy for Medline Ovid from 1946 to July 12, 2023.**

| | | |
|---|---|---|
| 1 | [peer-reviewed search date: 07/12/2023] | |
| 2 | [medline] | |
| 3 | [concept 1: MENA] | |
| 4 | (Algeria* or Bahrain* or Egypt* or Iran* or Iraq* or Israel* or Jordan* or Kuwait* or Lebanon* or Lebanese or Libya* or Morocc* or Oman or Omani or Oman's or Palestin* or Qatar* or Saudi Arabia* or Syria* or Tunisia* or Turkey* or United Arab Emirates or Emirat* or UAE or Western Sahara* or Yemen* or Middle East* or North* Africa* or MENA or "Middle East and North Africa" or Gaza* or Maghreb* or Persian Gulf).mp,jw. | 416652 |
| 5 | (Algiers or Manama or Cairo or Baghdad or Tehran or Amman or Kuwait City or Beirut or Tripoli or Rabat or Muscat or Ramallah or Jerusalem or Doha or Riyadh or Khartoum or Damascus or Tunis or Ankara or Abu Dhabi or Sana'a or Sanaa).mp. | 37116 |
| 6 | exp middle east/ or exp africa, northern/ | 200333 |
| 7 | (Turkiye* or Kurd* or near east* or Turkish or Istanbul).mp,jw. | 51343 |
| 8 | or/4-7 | 456675 |
| 9 | [concept 2: forced migrants] | |
| 10 | refugees/ or refugee camps/ | 13360 |
| 11 | (refugee* or asylum seeker* or migrant* or migration or immigra* or displac* or IDP or IDPs or "stateless person*" or "victims of persecution" or exiles or evacuees).mp. | 601424 |
| 12 | (asylee* or resettled or returnee*).mp. | 1441 |
| 13 | 0r/10-12 | 601805 |
| 14 | [concept 3: substance use or mental health disorders] | |
| 15 | [starting from the basis of Kumar, N., Janmohamed, K., Nyhan, K., Martins, S. S., Cerda, M., Hasin, D., Scott, J., Sarpong Frimpong, A., Pates, R., Ghandour, L. A., Wazaify, M., & Khoshnood, K. (2022). Substance use in relation to COVID-19: A scoping review. Addictive Behaviors, 127, 107213. https://doi.org/10.1016/j.addbeh.2021.107213 ] | |
| 16 | [substance use] | |
| 17 | exp substance-related disorders/ | 310855 |
| 18 | ((substance* adj3 (abus* or dependen* or disorder* or addict* or misus* or "use" or "user" or "users" or "usage" or "using" or "used" or withdrawal)) or addiction* or addict*).mp. | 260922 |
| 19 | ((opioid* or opiate* or heroin or amphetamine* or drug or drugs) adj3 (abus* or dependen* or disorder* or addict* or misus* or "use" or "user" or "users" or "usage" or "using" or "used")).mp. | 328542 |
| 20 | (OUD or IDU or PWID or SUD or SAT or MAT or OAT or AUD or PWUD or OTP or OST or PDM).mp. | 57226 |
| 21 | ("injection drug use" or "intravenous drug use").mp. | 6737 |
| 22 | methadone/ or opiate substitution treatment/ or exp buprenorphine/ | 20292 |
| 23 | harm reduction/ | 4130 |
| 24 | [redundant] | |
| 25 | exp smoking/ | 161200 |
| 26 | smoking cessation/ | 32748 |
| 27 | "tobacco use cessation"/ [check that quotation marks remain through launcher] | 1454 |
| 28 | vaping/ | 3558 |
| 29 | Electronic Nicotine Delivery Systems/ | 7856 |
| 30 | marijuana abuse/ or exp "marijuana use"/ | 12979 |
| 31 | exp tobacco products/ or smokers/ | 15479 |
| 32 | (smoking or smoker* or cigarette* or ecig* or e-cig* or tobacco or snuff or snus or cannabis or marijuana or vape or vaping or vaper or vapers or vaped or cigar*).mp. | 465778 |
| 33 | (alcohol* or "alcohol use" or alcohol abuse* or wine* or beer* or liquor* or spirits).mp. | 534626 |
| 34 | exp illicit drugs/ | 14063 |
| 35 | aerosol propellant*.mp. | 775 |
| 36 | ((sniff* or huff* or inhal*) adj2 aerosol*).mp. | 3194 |
| 37 | inhalant abuse.mp. | 468 |
| 38 | huffing.mp. | 94 |
| 39 | aliphatic nitrites.mp. | 5 |
| 40 | anabolic steroid*.mp. | 4762 |
| 41 | doping in sports/ and exp steroids/ | 931 |

(*Continued*)

**Table 1.** (Continued)

| 42 | performance-enhancing substances/ | 1285 |
|---|---|---|
| 43 | (phencyclidine or PCP or angel dust).mp. | 17553 |
| 44 | Benzodiazepines/ or exp Morphine Derivatives/ or exp Naloxone/ or exp Cocaine/ or exp Fentanyl/ or fentanyl.mp. or Narcotic Antagonists/ or Hallucinogens/ or Ketamine.mp. or Catha.mp. or exp Cannabis/ or "Ecstasy".mp. or MDMA.mp. or exp Methamphetamine/ or Mescaline.mp. or Opium/ or Tramadol/ or exp Narcotics/ | 268826 |
| 45 | benzodiazepine*.mp. | 50109 |
| 46 | cocaine.mp. | 47081 |
| 47 | (ecstasy or MDMA).mp. | 6613 |
| 48 | [redundant] | |
| 49 | N-Methyl-3,4-methylenedioxyamphetamine/ | 4253 |
| 50 | (GBH or GBL or gamma-hydroxybutyrate).mp. | 2178 |
| 51 | 4-Butyrolactone/ | 4530 |
| 52 | glue.mp. | 12838 |
| 53 | (hashish or heroin).mp. | 21647 |
| 54 | exp "hypnotics and sedatives"/ | 130569 |
| 55 | hypnotics.mp. | 34872 |
| 56 | inhalant*.mp. | 4797 |
| 57 | (LSD or Lysergic Acid Diethylamide).mp. | 9531 |
| 58 | mescaline/ | 1067 |
| 59 | mescaline.mp. | 1319 |
| 60 | exp amphetamines/ | 40260 |
| 61 | methamphetamine*.mp. | 16090 |
| 62 | (amphetamine* or speed).mp. | 252994 |
| 63 | methylxanthine*.mp. | 5832 |
| 64 | nicotine.mp. or nicotine chewing gum/ or nicotine/ | 56553 |
| 65 | (nitrous oxide or laughing gas).mp. | 24027 |
| 66 | ((OTC or over the counter) adj1 (drug* or medication*)).mp. | 3510 |
| 67 | exp nonprescription drugs/ | 6695 |
| 68 | (oxycodone or oxycontin).mp. | 5272 |
| 69 | paint thinner*.mp. | 124 |
| 70 | amyl nitrite.mp. | 918 |
| 71 | poppers.mp. | 483 |
| 72 | psilocybe.mp. | 211 |
| 73 | salvia divinorum.mp. | 233 |
| 74 | sedative*.mp. | 47365 |
| 75 | stimulant*.mp. | 47065 |
| 76 | tranquilizer*.mp. | 3199 |
| 77 | designer drugs/ | 1781 |
| 78 | overdose*.mp. | 31226 |
| 79 | narcotic*.mp. | 65858 |
| 80 | needle-exchange programs/ | 1994 |
| 81 | self medication/ | 4972 |
| 82 | opioid*.mp. | 147649 |
| 83 | ((medication-assisted or opiate-agonist or maintenance or replacement or substitution) adj1 (treatment* or therap*)).mp. | 116297 |
| 84 | [mental disorders] | |
| 85 | exp mental disorders/ or depression/ | 1529266 |
| 86 | mentally ill persons/ | 6431 |
| 87 | mental health/ | 61802 |

(*Continued*)

**Table 1.** (Continued)

| 88 | (mental health or depression or depressive or anxiety or PTSD or posttraumatic stress or posttraumatic stress or MMD or GAD or panic disorder* or adjustment disorder* or mental illness* or mood disorder* or behavioral health or behavioural health or psychological or stress or stressor or stressors or distress or well-being or well-being or resilience).mp. | 2560558 |
|---|---|---|
| 89 | exp nonprescription drugs/ or exp drug misuse/ | 23935 |
| 90 | (khat or qat or catha edulis or mairung* or miraa or hookah* or hooka* or narghile* or argila* or shisha or sheesha or waterpipe* or water pipe*).mp. or Catha/ | 4630 |
| 91 | or/17-90 | 5079215 |
| 92 | 8 and 13 and 91 | 3877 |
| 93 | limit 92 to yr = "2013 -Current" | 2311 |
| 94 | limit 93 to (arabic or english) | 2227 |

resolved by authors with expertise in the relevant field (MW, JT, KK). PRISMA flowchart diagram [30] depicts the screening and selection process once the review is finalized (Fig 1).

If there are any disagreements among the main reviewers over the selection, a third reviewer will make the final decision, or the two main reviewers will convene to discuss and make a final decision. The reviewers will then extract data from the selected studies using a predetermined form. The data extraction form will include details about the authors, publication year, sources of funding, countries, language, population characteristics (age and gender), sample size, type of intervention, prevalence, causes and risk factors of SUDs and MHDs, types of substances and mental illnesses, comparison tools, follow-up period, and other key findings. Relevant papers written in languages other than English and Arabic as well as grey literature such as conference papers and preprints will be listed in the S1 Chekclist.

### Risk of bias assessment

The risk of bias in the included studies will be assessed through multiple appraisal tools. Observational studies will be appraised with a revised version of JBI for systematic reviews of prevalence [31, 32], and intervention studies will be appraised by ROBINS-I [33, 34]. The studies pertaining to primary research question 1 (PQ1) will undergo appraisal, while those related to PQ2 and PQ3 as well as secondary research questions 1 (SQ1) and SQ2 will not be appraised. This is because the latter questions are of the scoping review type, where critical appraisal is not recommended.

Two reviewers will independently evaluate each study, and any discrepancies will be resolved through discussion and consensus. The tool assesses the risk of bias in several domains, including selection bias, performance bias, detection bias, attrition bias, reporting bias, and other sources of bias. Selection bias will be evaluated by assessing the adequacy of the randomization process and allocation concealment. Performance bias will be assessed by evaluating the blinding of participants and personnel. Detection bias will be evaluated by assessing the blinding of outcome assessors. Attrition bias will be evaluated by assessing the completeness of outcome data. Reporting bias will be evaluated by assessing the completeness of outcome reporting. Any other sources of bias will be evaluated based on the specific characteristics of each study. The risk of bias assessment will be used to inform the interpretation and synthesis of the study findings, and studies with high risk of bias will be considered in a sensitivity analysis.

Through sensitivity analysis, we will systematically exclude studies with methodological limitations or biases affecting internal validity, which have been appraised as high risk of bias using ROBINS-I. The specific methods and criteria for exclusion will be described to ensure

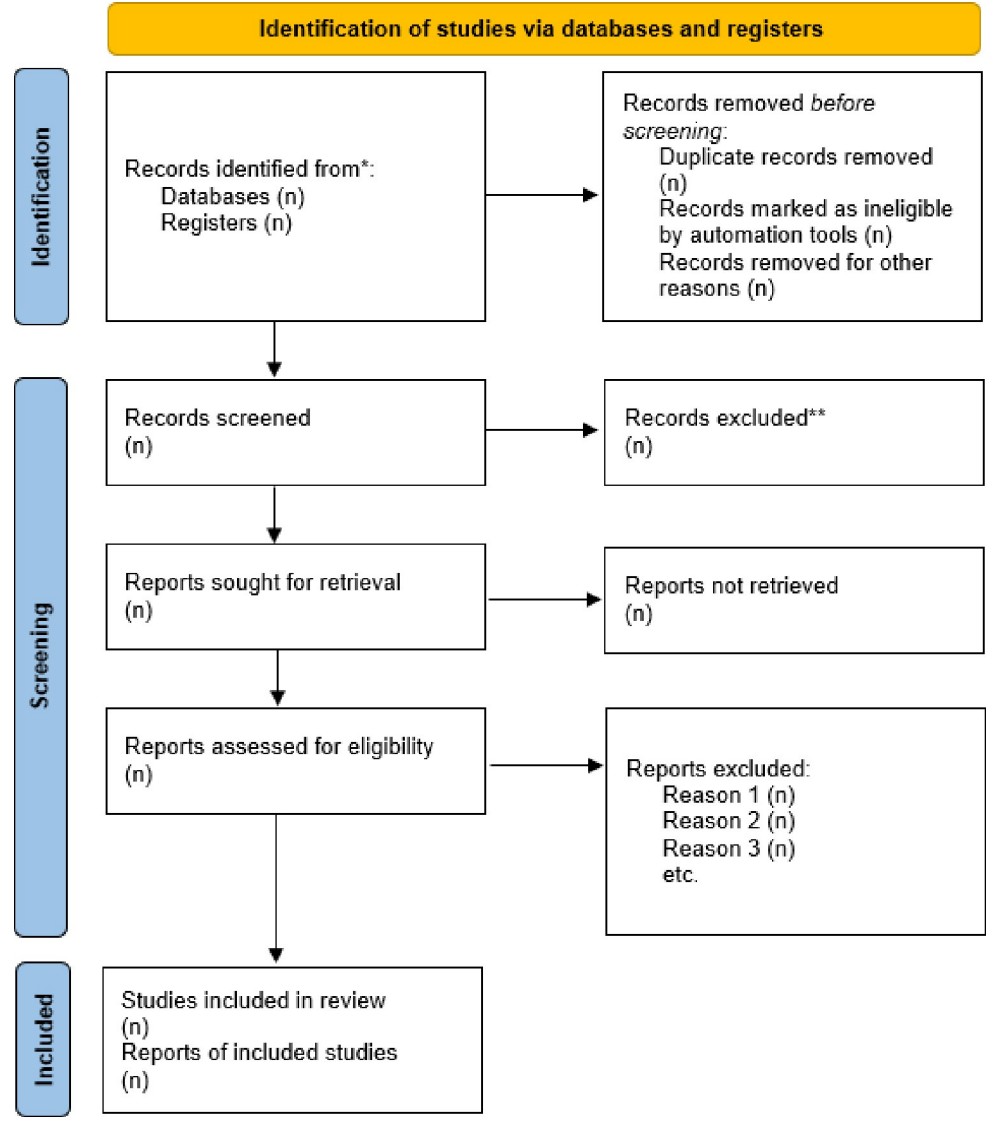

**Fig 1. PRISMA flow diagram.**

transparency. The impact of the sensitivity analysis on the conclusions and the strength of the evidence will be discussed, enhancing the reliability and validity of the review.

## Statistical analysis

The meta-analysis model will be chosen based on the study design. Given the expected variability among studies, a random-effects model will be considered due to its capacity to account for heterogeneity. If utilizing the 'meta' package, the 'metagen' function [35] will be applied. Alternatively, for the 'metafor' package, the 'rma' function [36] will be used. The selected function will be executed by inputting a structured data frame and defining relevant parameters including effect size, standard error, sample size, and study identifiers.

To quantify heterogeneity among the studies, $I^2$ statistic [37] will be employed to quantify the proportion of total variation in effect estimates that is due to heterogeneity rather than chance. Visual representations of study effect sizes and confidence intervals will be generated

through the creation of forest plots, using functions like 'forest()' or 'forest.rma()'. Potential publication bias will be explored and visualized using funnel plots and Egger's test [38] considering a $p$-value $< .05$. These analyses will be facilitated through dedicated functions available in the chosen R package.

Subgroup analyses will be conducted to find potential sources of heterogeneity by categorizing studies based on specific study characteristics. Variables anticipated to introduce heterogeneity will be considered for these analyses. Finally, for each variable, separate subgroups will be created, grouping studies with similar characteristics together. The meta-analysis model will then be applied within each subgroup to calculate the pooled effect estimate and associated confidence intervals. Comparing the effect estimates across different subgroups will allow for the identification of patterns or trends, revealing how each variable may impact the overall results. This process will provide valuable insights into the relative contribution of each characteristic to the observed heterogeneity.

## Data synthesis

In the data synthesis section of the systematic review, the primary studies included in the analysis will be analyzed and reported according to the research questions. We aim to investigate a comprehensive range of research questions to gain a thorough understanding of the topic under study. The PQ1uncovers international evidence, necessitating a systematic review approach. By following a systematic review methodology, we will conduct a rigorous and exhaustive literature search, ensuring the inclusion of all relevant studies and providing a robust synthesis and analysis of the findings [31].

Additionally, we recognize that PQ2, PQ3, SQ1, and SQ2 might benefit from a scoping review approach due to their exploratory and broad-scope nature. Employing a scoping review methodology for these secondary questions will enable us to map the available evidence, identify key concepts, and offer an overview of the literature without necessarily assessing the quality of individual studies. This approach will provide a comprehensive understanding of the existing research landscape related to these questions and facilitate the identification of knowledge gaps and potential areas for future research [31]. Given that the outcomes for PQ2/3 and SQ1/2 will be scoping review questions and no risk of bias assessments will be completed, there will be no evaluation conducted to determine the overall certainty or strength of the evidence.

## Discussion

The implications of this systematic review extend beyond research to practice and policy. By providing a more nuanced understanding of the prevalence and nature of SUDs and MHDs among forced migrants, the review may help inform the development of targeted interventions and services tailored to the specific needs of this population. For example, the review may suggest the need for culturally sensitive and linguistically appropriate services that take into account the unique experiences and challenges faced by forced migrants. Additionally, the review may help to identify gaps in existing policies and services, highlighting areas where further investment and resource allocation are needed to better support forced migrants with SUDs and MHDs.

Finally, it is important to note that this review is just one step towards improving our understanding of SUDs and MHDs among forced migrants. Ongoing research and investment in this area is crucial to better understand the complex interplay between forced migration, substance use, and mental health. Ultimately, we hope this research helps to improve the health

and well-being of forced migrants, a population that is often underserved and marginalized in the context of global health.

## Limitations

While we will endeavor to find all original articles aligned with our eligibility criteria through bibliographic database searches and citation chaining, we might miss some relevant studies. In addition, we may not be able to make policy recommendations due to the lack of high-quality studies. Last but not least, some authors we contact may not respond, and we may thus miss including those articles if the published documents lack important details.

## Supporting information

**S1 Checklist. PRISMA-P (Preferred Reporting Items for Systematic review and Meta-Analysis Protocols) 2015 checklist: Recommended items to address in a systematic review protocol\*.**
(DOC)

## Acknowledgments

The authors would like to thank Tom Mead for his valuable contributions in peer-reviewing the search strategy employed in this study.

## Author Contributions

**Conceptualization:** Maryam Kazemitabar.

**Methodology:** Kate Nyhan.

**Resources:** Kate Nyhan.

**Validation:** Kate Nyhan.

**Writing – original draft:** Maryam Kazemitabar, Natalie Makableh, Rebecca Minahan-Rowley, Malak Ali.

**Writing – review & editing:** Maryam Kazemitabar, Kate Nyhan, Natalie Makableh, Rebecca Minahan-Rowley, Malak Ali, Mayyada Wazaify, Jeanette Tetrault, Kaveh Khoshnood.

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
