## [Decision Letter · Decision Letter 0]

29 Jun 2023

PONE-D-23-12703

Epidemiology of substance use and mental health disorders among forced migrants displaced from the MENA region: a systematic review protocol

PLOS ONE

Dear Dr. Kazemitabar,

Thank you for submitting your manuscript to PLOS ONE. After careful consideration, we feel that it has merit but does not fully meet PLOS ONE’s publication criteria as it currently stands. Therefore, we invite you to submit a revised version of the manuscript that addresses the points raised during the review process.

We look forward to receiving your revised manuscript.

Kind regards,

Amin Nakhostin-Ansari

Academic Editor

PLOS ONE

Journal Requirements:

Additional Editor Comments:

Thank you for submitting your manuscript to PLOS ONE. I greatly appreciate the contribution of your study and recognize its value in the field. Two experts have reviewed your manuscript and have provided constructive feedback. Based on their evaluations, we have identified specific methodological aspects that require further attention. Please carefully address the reviewers' comments, as their insights will greatly contribute to enhancing the quality of your protocol and the subsequent review.

Reviewers' comments:

Reviewer's Responses to Questions

**Comments to the Author**

1. Does the manuscript provide a valid rationale for the proposed study, with clearly identified and justified research questions?

Reviewer #1: Yes

Reviewer #2: Partly

2. Is the protocol technically sound and planned in a manner that will lead to a meaningful outcome and allow testing the stated hypotheses?

Reviewer #1: Yes

Reviewer #2: No

3. Is the methodology feasible and described in sufficient detail to allow the work to be replicable?

Reviewer #1: Yes

Reviewer #2: Yes

4. Have the authors described where all data underlying the findings will be made available when the study is complete?

Reviewer #1: No

Reviewer #2: Yes

5. Is the manuscript presented in an intelligible fashion and written in standard English?

Reviewer #1: Yes

Reviewer #2: Yes

6. Review Comments to the Author

You may also provide optional suggestions and comments to authors that they might find helpful in planning their study.

Reviewer #1: This protocol outlines a well-designed study and important study on the epidemiology of substance use and mental health disorders among MENA forced migrants. The authors motivate the study well and prose a technically sound study that should achieve their research aims. Notwithstanding these considerable strengths, the submission could be strengthened through several additional considerations:

1. Introduction:

a. It would be useful to add a few lines articulating exactly why the MENA population warrants further study, other than there being a literature gap. After all, if the state of the literature is so poor, then why conduct a systematic literature review and not something more exploratory, such as a scoping study? It would be worth providing a short review of the sizable literature on the mental health needs and experiences of the MENA population and to position this study in relation to other available reviews.

2. Inclusion/exclusion criteria:

a. The authors’ definition of “forced migrants displaced from the MENA region” could be sharpened. Taken at face value, this wording suggests that the authors will only include studies on people that have been displaced from the region entirely, which would exclude the steep majority, who have been displaced within the region. From the search terms, I don’t believe this is the authors’ intention, but it would be worth stating in the inclusion/exclusion criteria section exactly what they intend. For instance: studies focused on people from the MENA region who have been forcibly displaced by conflict, persecution, and/or natural disaster, including international displacement as well as internal displacement. Relatedly, the authors’ definition of the MENA region is a little fuzzy, since UNOCHA responds to both places producing displacement (e.g., Syria) and places receiving displaced people (e.g., Jordan). Often, for example, Turkey is not included in definitions of the MENA region (for instance, in OHCHR), but UNOCHA includes it because of its Syrian response. Some related questions: Is there a minimum/maximum length of time for the length of displacement that the authors will include? For instance, will they include resettled refugees, who are technically no longer displaced? Will they include people who fled a natural disaster for a month, only to return? Will they include returnees more generally?

b. Will the authors include studies that include participants from other regions, or only studies that focus exclusively on migrants from the MENA region?

c. The authors justify the 2013-2023 period with a parenthetical reference to the Arab Spring, but it’s unclear how this fits exactly. The Conventional periodization of the Arab Spring is 2010-2012, so is the goal to include only studies that followed the Arab Spring? Why not also include 2010-2012? As an aside, the introduction section attributes displacement to the Arab Spring protest, rather than to the state crackdowns, coups, and war that followed. I would consider reframing.

d. Will the authors include qualitative, quantitative studies, and mixed-methods studies, or only quantitative studies?

3. Search strategy:

a. Concept 1: Consider adding: Türkiye (official name), Kurdistan, and Near East

b. Concept 2: Depending on your responses to the above, consider adding asylee, resettled, and returnee

c. Concept 3: Consider adding regionally specific substances, such as khat/qat, hookah/narghile/argila/shisha/waterpipe

d. Consider adding search terms in Arabic

Thank you for the opportunity to read and review this protocol.

Reviewer #2: Thank you for inviting me to review the protocol “Epidemiology of substance use and mental health disorders among forced migrants 2 displaced from the MENA region: a systematic review protocol”. Overall, I believe that the review topic itself has merit, but this manuscript would benefit from major revisions, and perhaps even the assistance of an editor for help with writing as there is awkward phrasing throughout the paper. I have provided my comments as both a topic expert (migrant health) and as a systematic review methodologist. I hope the authors find my comments useful in preparing their next version.

ABSTRACT

The abstract requires substantial revisions – I suggest that the authors review the PRISMA 2020 for Abstracts Checklist (http://prisma-statement.org/Extensions/Abstracts) to help with the overall reporting and structure. Importantly, the authors should provide an explicit statement of the main objective(s) or question(s) the review addresses in the background and define all acronyms at their first mention (e.g., “SU”, “MH”, “MENA”). The Methods should describe the inclusion criteria and the planned approach for synthesis.

INTRODUCTION

Major:

- Please review the UNHCR statistics and verify if more recent stats are available (2022? 2021?).

- Many statements are provided without in-text citations (e.g., Lines 51-52; 63-71). Please provide supporting references.

Minor:

- Lines 77 and 78 are repetitive.

METHODS

Major:

Given the broad nature of the review questions presented by the authors, and the descriptive planned synthesis and presentation of the results, I strongly recommend the authors to consider whether a scoping review is more appropriate review choice. The authors should justify their choice in their protocol. There are resources available on making this decision, as well as guidance for conduct and reporting:

- Munn, Z., Peters, M. D., Stern, C., Tufanaru, C., McArthur, A., & Aromataris, E. (2018). Systematic review or scoping review? Guidance for authors when choosing between a systematic or scoping review approach. BMC medical research methodology, 18, 1-7. https://bmcmedresmethodol.biomedcentral.com/articles/10.1186/s12874-018-0611-x

- JBI MANUAL FOR EVIDENCE SYNTHESIS: SCOPING REVIEWS CHAPTER: https://jbi.global/scoping-review-network/resources

- Tricco, AC, Lillie, E, Zarin, W, O'Brien, KK, Colquhoun, H, Levac, D, Moher, D, Peters, MD, Horsley, T, Weeks, L, Hempel, S et al. PRISMA extension for scoping reviews (PRISMA-ScR): checklist and explanation. Ann Intern Med. 2018,169(7):467-473. doi:10.7326/M18-0850. http://www.prisma-statement.org/Extensions/ScopingReviews

Under “Participants”, the authors should define the term “forced migrants”. For example, will the authors include migrants who cross a border (refugees, asylum seekers) and those who don’t (IDPs?). What types of migrant populations would be excluded (e.g., economic migrants, international students, other?).

If the authors choose to conduct a systematic review, then they must specify interventions and outcomes eligible for inclusion.

Under “Risk of Bias Assessment” – the authors have selected the Cochrane RoB 2.0 tool to assess intervention studies, however this tool it specifically designed for RCTs and does not adequately assess other non-randomized study designs. The authors should consider ROBINS-I. Given that the studies addressing review question #1 are likely to be observational and reporting prevalence, the authors should also consider whether the JBI critical appraisal checklist for prevalence studies would be beneficial. Additional guidance for reporting such evidence is available:

- Munn Z, Moola S, Lisy K, Riitano D, Tufanaru C. Methodological guidance for systematic reviews of observational epidemiological studies reporting prevalence and incidence data. Int J Evid Based Healthc. 2015;13(3):147–153.

Under “Data synthesis” it’s not clear why the authors have not described a synthesis approach for the pooling of prevalence estimates, which would be expected in a systematic review addressing question #1. The charting approach described by the authors would be better suited to a scoping review.

The authors previously described that “studies with high risk of bias will be considered in a sensitivity analysis”, but the methods for this analysis are not described in this section.

Given the explicit decision not to pool outcomes, the authors will find it challenging to conduct GRADE assessments, as there may be a large number of single-study estimates. Additionally, the authors have not explicitly defined any outcomes in their eligibility criteria or analysis plan, and so it is unclear what types of outcomes will be assessed using GRADE. Notably, there is currently no GRADE guidance available for the assessment of the certainty of evidence of prevalence data. The authors should describe how they plan to assess the evidence for review question 1.

Minor:

- Line 97 – it is not necessary to describe PROSPERO and cite Booth et al., your reference should include the citation information necessary to locate your registration.

- There is a word missing, Line 99: “The current ___ uses the framework…”

- The headings “study selection criteria” and “inclusion and exclusion criteria” are redundant – The PRISMSA-P Checklist uses the language “Eligibility criteria” – the authors should review PRISMA-P and ensure that they are adequately reporting all items.

- Under “screening”, the authors should specify that screening will be done in duplicate

7. PLOS authors have the option to publish the peer review history of their article (what does this mean?). If published, this will include your full peer review and any attached files.

Reviewer #1: **Yes: **Cyril Bennouna

Reviewer #2: No

---

## [Author Response · Author response to Decision Letter 0]

3 Aug 2023

Dear Reviewers,

I extend my sincere gratitude to the reviewers for their valuable comments and feedback, which greatly improved my revised manuscript. Your input was instrumental in enhancing the quality of our work, and I appreciate the time and effort you dedicated to the review process. Thank you for your invaluable contributions to this research.

Sincerely,

Maryam Kazemitabar

Please see our responses to the reviewers’ comments below. My responses to the reviewers' comments are in bold black and when I directly referred to the manuscript contents in bold blue.

Reviewer #1: This protocol outlines a well-designed study and important study on the epidemiology of substance use and mental health disorders among MENA forced migrants. The authors motivate the study well and prose a technically sound study that should achieve their research aims. Notwithstanding these considerable strengths, the submission could be strengthened through several additional considerations:

1. Introduction:

a. It would be useful to add a few lines articulating exactly why the MENA population warrants further study, other than there being a literature gap. After all, if the state of the literature is so poor, then why conduct a systematic literature review and not something more exploratory, such as a scoping study? It would be worth providing a short review of the sizable literature on the mental health needs and experiences of the MENA population and to position this study in relation to other available reviews.

Response to the reviewer’s comment: Thank you for your helpful comment. I revised it, you can see the changes on pages 3 and 4 lines 77-83: “Studying the epidemiology of SUDs and MHDs among forced migrants from the MENA region is crucial due to their unique challenges and vulnerabilities. There is a substantial body of existing research on the topic of SUDs [12, 13] and especially MHDs [14-18] from the MENA region. However, the availability of systematic reviews that comprehensively collect and analyze this data is limited. Conducting a systematic literature review will help bridge this gap by providing a comprehensive synthesis of the existing research, allowing for a more thorough understanding of the epidemiology of SUDs and MHDs among this population.”

2. Inclusion/exclusion criteria:

a. The authors’ definition of “forced migrants displaced from the MENA region” could be sharpened. Taken at face value, this wording suggests that the authors will only include studies on people that have been displaced from the region entirely, which would exclude the steep majority, who have been displaced within the region. From the search terms, I don’t believe this is the authors’ intention, but it would be worth stating in the inclusion/exclusion criteria section exactly what they intend. For instance: studies focused on people from the MENA region who have been forcibly displaced by conflict, persecution, and/or natural disaster, including international displacement as well as internal displacement. 

Response to the reviewer’s comment: Thank you for your valuable feedback. We intended to include both internationally and internal displacement, so we included this sentence per your suggestion. Please see page 6, lines 140-142: “Studies focused on people from the MENA region who have been forcibly displaced by conflict, persecution, and/or natural disaster, including international displacement as well as internal displacement will be included.”.

Relatedly, the authors’ definition of the MENA region is a little fuzzy, since UNOCHA responds to both places producing displacement (e.g., Syria) and places receiving displaced people (e.g., Jordan). Often, for example, Turkey is not included in definitions of the MENA region (for instance, in OHCHR), but UNOCHA includes it because of its Syrian response.

Response to the reviewer’s comment: Thank you for your insightful comment. We used United Nations Human Resources Office of the High Commissioner (HROHC) classification for included countries in MENA region plus Türkiye (MENAT).

Some related questions: Is there a minimum/maximum length of time for the length of displacement that the authors will include? For instance, will they include resettled refugees, who are technically no longer displaced? Will they include people who fled a natural disaster for a month, only to return? Will they include returnees more generally?

Response to the reviewer’s comment: There is no specific length of time for displacement in this study. All refugees resettled or not resettled will be included since even after resettlement they might struggle with SUDs and/or MHDs, we decided to include them all.

b. Will the authors include studies that include participants from other regions, or only studies that focus exclusively on migrants from the MENA region?

Response to the reviewer’s comment: Only studies on forced migrants from the MENA region. Expanding the inclusion criteria to encompass forced migrants from other regions would significantly increase the number of extracted articles, making the process of data extraction and synthesis unmanageable. Furthermore, there already exists a substantial body of research focusing on MHDs among forced migrants specifically from the MENA region.

c. The authors justify the 2013-2023 period with a parenthetical reference to the Arab Spring, but it’s unclear how this fits exactly. The Conventional periodization of the Arab Spring is 2010-2012, so is the goal to include only studies that followed the Arab Spring? Why not also include 2010-2012? As an aside, the introduction section attributes displacement to the Arab Spring protest, rather than to the state crackdowns, coups, and war that followed. I would consider reframing.

Response to the reviewer’s comment: We aim to include the studies conducted in the last ten years not more to make the process of data extraction and synthesis manageable. The introduction section is reframed per your suggestion. Please see page 3, lines 59-62: “The Arab Spring protest and the subsequent state crackdowns, coups, and wars that followed the initial protests played a significant role in the displacement such that caused more than 3.5 million people to get internally displaced; ever since the numbers have even increased to more than three times.”

d. Will the authors include qualitative, quantitative studies, and mixed-methods studies, or only quantitative studies?

Response to the reviewer’s comment: All qualitative, quantitative studies, and mixed-methods studies will be included. We added this information to the methods section. Please see page 6, lines 133-134: “The systematic review will include all qualitative, quantitative studies, and mixed-methods studies focusing on SUDs and/or MHDs among forced migrants displaced from the MENA region.”

3. Search strategy:

a. Concept 1: Consider adding: Türkiye (official name), Kurdistan, and Near East

b. Concept 2: Depending on your responses to the above, consider adding asylee, resettled, and returnee

c. Concept 3: Consider adding regionally specific substances, such as khat/qat, hookah/narghile/argila/shisha/waterpiped. Consider adding search terms in Arabic

Response to the reviewer’s comment : Thank you for these suggestions about additional search terms. We have added them all, as well as demonyms for Turkish and Kurdish people, and some additional synonyms/spellings for the regionally specific substances.

Regarding the suggestion of adding search terms in Arabic: we considered this approach, but we decided not to implement it. We are confident that we can nevertheless expect to identify relevant Arabic-language papers with the current approach, through three pathways. First, many Arabic-language papers in the databases we are searching have English-language abstracts, through which our search strategy will retrieve them. Second, Arabic-language papers in the databases which we are searching with controlled vocabulary will be retrieved by our use of subject headings and our use of the Ovid multipurpose field (as opposed to the Ovid textword field). Third, papers (in any language) in journals that are not indexed in any of the databases which we searched can still be identified in this project through citation chaining, which we will conduct with the tool Citation Chaser (Haddaway 2022), using the bibliographic database Lens, which includes publication metadata from the database/dataset OpenAlex (https://about.lens.org/release-8-5/), which is recognized as having particularly good coverage of open access journals based in LMIC countries (Khanna 2022). Via these pathways, we hope to identify relevant papers in Arabic, even if they are from journals not indexed in any of the bibliographic databases we are searching. What’s more, we intend to create a supplemental table of relevant articles in other languages – Persian, French, Turkish, any other language – that we identify during the screening process, for use by other researchers studying this topic in the future. 

Khanna, S., Ball, J., Alperin, J. P., & Willinsky, J. (2022). Recalibrating the Scope of Scholarly Publishing: A Modest Step in a Vast Decolonization Process. Quantitative Science Studies, 1–43. https://doi.org/10.1162/qss_a_00228

Haddaway, N. R., Grainger, M. J., & Gray, C. T. (2022). Citationchaser: A tool for transparent and efficient forward and backward citation chasing in systematic searching. Research Synthesis Methods, n/a(n/a). https://doi.org/10.1002/jrsm.1563

Reviewer #2: Thank you for inviting me to review the protocol “Epidemiology of substance use and mental health disorders among forced migrants 2 displaced from the MENA region: a systematic review protocol”. Overall, I believe that the review topic itself has merit, but this manuscript would benefit from major revisions, and perhaps even the assistance of an editor for help with writing as there is awkward phrasing throughout the paper. I have provided my comments as both a topic expert (migrant health) and as a systematic review methodologist. I hope the authors find my comments useful in preparing their next version.

ABSTRACT

The abstract requires substantial revisions – I suggest that the authors review the PRISMA 2020 for Abstracts Checklist (http://prisma-statement.org/Extensions/Abstracts) to help with the overall reporting and structure. Importantly, the authors should provide an explicit statement of the main objective(s) or question(s) the review addresses in the background and define all acronyms at their first mention (e.g., “SU”, “MH”, “MENA”). The Methods should describe the inclusion criteria and the planned approach for synthesis.

Response to the reviewer’s comment: Thank you for your constructive feedback. The Abstract is revised and improved now.

INTRODUCTION

Major:

- Please review the UNHCR statistics and verify if more recent stats are available (2022? 2021?).

Response to the reviewer’s comment: Data is updated from UNHRC.

- Many statements are provided without in-text citations (e.g., Lines 51-52; 63-71). Please provide supporting references.

Response to the reviewer’s comment: References added.

Minor:

- Lines 77 and 78 are repetitive.

Response to the reviewer’s comment: Thank you for noticing that. This sentence is modified.

METHODS

Major:

Given the broad nature of the review questions presented by the authors, and the descriptive planned synthesis and presentation of the results, I strongly recommend the authors to consider whether a scoping review is more appropriate review choice. The authors should justify their choice in their protocol. There are resources available on making this decision, as well as guidance for conduct and reporting:

- Munn, Z., Peters, M. D., Stern, C., Tufanaru, C., McArthur, A., & Aromataris, E. (2018). Systematic review or scoping review? Guidance for authors when choosing between a systematic or scoping review approach. BMC medical research methodology, 18, 1-7. https://bmcmedresmethodol.biomedcentral.com/articles/10.1186/s12874-018-0611-x- JBI MANUAL FOR EVIDENCE SYNTHESIS: SCOPING REVIEWS CHAPTER: https://jbi.global/scoping-review-network/resources

- Tricco, AC, Lillie, E, Zarin, W, O'Brien, KK, Colquhoun, H, Levac, D, Moher, D, Peters, MD, Horsley, T, Weeks, L, Hempel, S et al. PRISMA extension for scoping reviews (PRISMA-ScR): checklist and explanation. Ann Intern Med. 2018,169(7):467-473. doi:10.7326/M18-0850. http://www.prisma-statement.org/Extensions/ScopingReviews

Response to the reviewer’s comment: We appreciate the reviewer's thoughtful consideration of the review questions and the recommended resources. We have thoroughly reviewed the suggestions and have reevaluated the appropriateness of the review type for each question.

After careful consideration, we agree that the primary question (PQ1) has a broad nature and requires a systematic review approach. We will conduct a comprehensive and methodical search of the literature to identify all relevant studies, followed by a rigorous synthesis and analysis of the findings. A systematic review will allow us to address PQ1 with a high level of evidence and ensure a robust and reliable conclusion.

Regarding the secondary questions, SQ1 and SQ2, and primary question 2 (PQ2) we acknowledge that they might benefit from a scoping review approach due to their exploratory and broad scope nature. A scoping review will enable us to map the available evidence, identify key concepts, and provide an overview of the literature without necessarily assessing the quality of individual studies. This approach will help us gain a better understanding of the existing research landscape related to these questions.

In our protocol, we will explicitly specify the review type for each question and provide a clear justification for our choice. We will also refer to the recommended resources, such as the Munn et al. (2018) paper on choosing between systematic and scoping reviews and the PRISMA-ScR checklist by Tricco et al. (2018), to ensure that our review follows best practices for conduct and reporting.

Please see page 14, lines 240-252: “We aim to investigate a comprehensive range of research questions to gain a thorough understanding of the topic under study. The primary question (PQ1) has been identified as having a broad nature, necessitating a systematic review approach. By following a systematic review methodology, we will conduct a rigorous and exhaustive search of the literature, ensuring the inclusion of all relevant studies and providing a robust synthesis and analysis of the findings.

Additionally, we recognize that PQ2, SQ1, and SQ2 might benefit from a scoping review approach due to their exploratory and broad scope nature. Employing a scoping review methodology for these secondary questions will enable us to map the available evidence, identify key concepts, and offer an overview of the literature without necessarily assessing the quality of individual studies. This approach will provide a comprehensive understanding of the existing research landscape related to these questions and facilitate the identification of knowledge gaps and potential areas for future research.”.

Under “Participants”, the authors should define the term “forced migrants”. For example, will the authors include migrants who cross a border (refugees, asylum seekers) and those who don’t (IDPs?). What types of migrant populations would be excluded (e.g., economic migrants, international students, other?).

Response to the reviewer’s comment: The definition added per your suggestion.

If the authors choose to conduct a systematic review, then they must specify interventions and outcomes eligible for inclusion.

Response to the reviewer’s comment: Thank you for your comment. This systematic review is a systematic review of prevalence and incidence. Including interventions and outcomes is not applicable; it is more applicable for Cochrane-style systematic review of interventions.

Under “Risk of Bias Assessment” – the authors have selected the Cochrane RoB 2.0 tool to assess intervention studies, however this tool it specifically designed for RCTs and does not adequately assess other non-randomized study designs. The authors should consider ROBINS-I. 

Response to the reviewer’s comment: Thank you for your suggestion. Edited!

Given that the studies addressing review question #1 are likely to be observational and reporting prevalence, the authors should also consider whether the JBI critical appraisal checklist for prevalence studies would be beneficial. Additional guidance for reporting such evidence is available:

- Munn Z, Moola S, Lisy K, Riitano D, Tufanaru C. Methodological guidance for systematic reviews of observational epidemiological studies reporting prevalence and incidence data. Int J Evid Based Healthc. 2015;13(3):147–153.

Response to the reviewer’s comment: Thank you for your helpful comment. We will use the JBI critical appraisal checklist for prevalence study that you mentioned. The manuscript’s content edited accordingly.

Under “Data synthesis” it’s not clear why the authors have not described a synthesis approach for the pooling of prevalence estimates, which would be expected in a systematic review addressing question #1. The charting approach described by the authors would be better suited to a scoping review.

Response to the reviewer’s comment: The approach to synthesizing prevalence studies is added to the manuscript per your suggestion: “The ‘best evidence synthesis’ will be used to synthesizing prevalence estimates. In implementing a best evidence synthesis approach, this study will prioritize higher-quality studies or those with more robust methodology to guide the synthesis process [29]. Greater weight will be assigned to studies with larger sample sizes, rigorous study designs, or lower risk of bias when summarizing prevalence estimates. By employing this approach, we aim to provide a more reliable and valid synthesis of prevalence data that will be informed by studies of the highest quality and methodological rigor.”

29- Slavin, R.E., Best-Evidence Synthesis: An Alternative to Meta-Analytic and Traditional Reviews. Educational Researcher, 1986. 15(9): p. 5-11.

The authors previously described that “studies with high risk of bias will be considered in a sensitivity analysis”, but the methods for this analysis are not described in this section.

Response to the reviewer’s comment: We appreciate the reviewer's comment. We added a description about methods for sensitivity analysis.

Given the explicit decision not to pool outcomes, the authors will find it challenging to conduct GRADE assessments, as there may be a large number of single-study estimates. Additionally, the authors have not explicitly defined any outcomes in their eligibility criteria or analysis plan, and so it is unclear what types of outcomes will be assessed using GRADE. Notably, there is currently no GRADE guidance available for the assessment of the certainty of evidence of prevalence data. The authors should describe how they plan to assess the evidence for review question 1.

Response to the reviewer’s comment: Thank you for your comment. We will use ‘best evidence synthesis’ to synthesizing prevalence estimates as mentioned above.

Minor:

- Line 97 – it is not necessary to describe PROSPERO and cite Booth et al., your reference should include the citation information necessary to locate your registration.

Response to the reviewer’s comment: We have included registration ID under the “Design of the systematic review” section. Citation for PROSPERO removed.

- There is a word missing, Line 99: “The current ___ uses the framework…”

Response to the reviewer’s comment: Edited! Thank you.

- The headings “study selection criteria” and “inclusion and exclusion criteria” are redundant – The PRISMSA-P Checklist uses the language “Eligibility criteria” – the authors should review PRISMA-P and ensure that they are adequately reporting all items.

Response to the reviewer’s comment: Edited! Thank you.

- Under “screening”, the authors should specify that screening will be done in duplicate

Response to the reviewer’s comment: Added.

---

## [Decision Letter · Decision Letter 1]

15 Aug 2023

PONE-D-23-12703R1Epidemiology of substance use and mental health disorders among forced migrants displaced from the MENAT region: a systematic review protocolPLOS ONE

Dear Dr. Kazemitabar,

Thank you for submitting your manuscript to PLOS ONE. After careful consideration, we feel that it has merit but does not fully meet PLOS ONE’s publication criteria as it currently stands. Therefore, we invite you to submit a revised version of the manuscript that addresses the points raised during the review process.

We look forward to receiving your revised manuscript.

Kind regards,

Amin Nakhostin-Ansari

Academic Editor

PLOS ONE

Reviewers' comments:

Reviewer's Responses to Questions

**Comments to the Author**

1. Does the manuscript provide a valid rationale for the proposed study, with clearly identified and justified research questions?

Reviewer #1: Yes

Reviewer #2: Yes

2. Is the protocol technically sound and planned in a manner that will lead to a meaningful outcome and allow testing the stated hypotheses?

Reviewer #1: Yes

Reviewer #2: Partly

3. Is the methodology feasible and described in sufficient detail to allow the work to be replicable?

Reviewer #1: Yes

Reviewer #2: No

4. Have the authors described where all data underlying the findings will be made available when the study is complete?

Reviewer #1: No

Reviewer #2: Yes

5. Is the manuscript presented in an intelligible fashion and written in standard English?

Reviewer #1: Yes

Reviewer #2: Yes

6. Review Comments to the Author

You may also provide optional suggestions and comments to authors that they might find helpful in planning their study.

Reviewer #1: The authors have thoroughly addressed my comments and produced an improved submission. I have two small remaining notes: First, I find the justification for the systematic review for PQ1 somewhat confusing. In response to Reviewer 2, the authors note that "After careful consideration, we agree that the primary question (PQ1) has a broad nature and requires a systematic review approach." They then write in the body of the text that "The

primary question (PQ1) has been identified as having a broad nature, necessitating a systematic review approach." This is further confused by the fact that the authors also justify a *scoping* review for PQ2, SQ1, and SQ2 on the basis of their "broad scope nature." I disagree that "having a broad nature" is what warrants a systematic review, for PQ1, and I believe Reviewer 2 actually suggested a *scoping* review (rather than a systematic review) specifically "given the broad nature of the review questions." This is in keeping with the Munn et al. article on deciding between a systematic and scoping review, where they note that "A key difference between scoping reviews and systematic reviews is that in terms of a review question, a scoping review will have a broader “scope” than traditional systematic reviews..." That said, I agree with the authors that a systematic review is appropriate for PQ1—given the question's focus on synthesizing evidence of prevalence—but I would suggest articulating the justification more clearly. For example, using the same Munn et al. article, the authors could argue that PQ1 meets the systematic review indication for "uncover[ing] the international evidence." My second small comment is that some of the edits have introduced new grammatical errors (e.g. page 3, line 75), so I would recommend a comprehensive copy edit before publishing.

Reviewer #2: Thank you for addressing my previous comments and for inviting me to review the revised paper. I appreciate the efforts the authors have made in their protocol. Below, I have a few additional comments that should be addressed prior to publication.

1) The abstract states that you will use the Newcastle-Ottawa Scale and Cochrane RoB, but the methods have been revised to the JBI checklist for prevalence studies an d ROBINS-I. This needs to be consistent.

2) More details are needed regarding how you plan to analyze the data for PQ1. For example, do you plan to conduct a meta-analysis of prevalence estimates? If yes, will you use a random or fixed effects model? Will these estimates be stratified by MH condition? Will you conduct any assessments for publication bias? You do describe the best evidence synthesis approach to weight studies in your analyses, but it is still necessary to describe the analytic approach. I would consider reviewing the paper by Blackmore et al. below as an example of the type of analyses that could be possible. If the authors choose not to conduct a meta-analysis (or other regression analyses), this choice needs to be justified in the protocol.

Citation: Blackmore R, Boyle JA, Fazel M, Ranasinha S, Gray KM, Fitzgerald G, et al. (2020) The prevalence of mental illness in refugees and asylum seekers: A systematic review and meta-analysis. PLoS Med 17(9): e1003337. https://doi.org/10.1371/journal.pmed.1003337

3) The authors have still not addressed the issue with planning to use GRADE methods. Given that the outcomes for PQ2 and SQ1/2 will be scoping review questions and no risk of bias assessments will be complete, no GRADE assessment will be possible for these questions. Additionally, there are no current methods for GRADE assessments for prevalence data (PQ1). My suggestion would be for the authors to acknowledge these limitations and state that no assessment for the overall certainty/strength of the evidence will be performed.

4) In response to reviewers #1's comments, the authors described "papers (in any language) in journals that are not indexed in any of the databases which we searched can still be identified in this project through citation chaining, which we will conduct with the tool Citation Chaser (Haddaway 2022), using the bibliographic database Lens, which includes publication metadata from the database/dataset OpenAlex (https://about.lens.org/release-8-5/), which is recognized as having particularly good coverage of open access journals based in LMIC countries (Khanna 2022)."

These details (such as your planned use of the tool citation chaser) should be described in the protocol and referenced appropriately. You should list the software you plan to use for your analysis (e.g., RevMan, STATA, R) and any planned visualizations.

7. PLOS authors have the option to publish the peer review history of their article (what does this mean?). If published, this will include your full peer review and any attached files.

Reviewer #1: **Yes: **Cyril Bennouna

Reviewer #2: No

---

## [Author Response · Author response to Decision Letter 1]

15 Aug 2023

28th July, 2023

Dear Editorial Board,

We extend our sincere appreciation for the valuable and insightful feedback provided by the reviewers regarding our manuscript titled "Epidemiology of substance use and mental health disorders among forced migrants displaced from the MENA region: a systematic review protocol". We are pleased to submit the revised version of our manuscript to PLOS ONE journal.

The reviewers' comments have significantly contributed to the enhancement of our manuscript. Below, we have addressed each comment raised and incorporated the necessary revisions accordingly. Furthermore, we affirm that the present work has not been published in any journal previously, nor is it under consideration for publication elsewhere.

Sincerely,

Maryam Kazemitabar, Ph.D.

Department of Internal Medicine, Yale University

Email: maryam.kazemitabar@yale.edu

Please see our responses to the reviewers’ comments below.

Reviewer #1: This protocol outlines a well-designed study and important study on the epidemiology of substance use and mental health disorders among MENA forced migrants. The authors motivate the study well and prose a technically sound study that should achieve their research aims. Notwithstanding these considerable strengths, the submission could be strengthened through several additional considerations:

1. Introduction:

a. It would be useful to add a few lines articulating exactly why the MENA population warrants further study, other than there being a literature gap. After all, if the state of the literature is so poor, then why conduct a systematic literature review and not something more exploratory, such as a scoping study? It would be worth providing a short review of the sizable literature on the mental health needs and experiences of the MENA population and to position this study in relation to other available reviews.

Response to the reviewer’s comment: Thank you for your helpful comment. I revised it, you can see the changes on pages 3 and 4 lines 77-83: “Studying the epidemiology of SUDs and MHDs among forced migrants from the MENA region is crucial due to their unique challenges and vulnerabilities. There is a substantial body of existing research on the topic of SUDs [12, 13] and especially MHDs [14-18] from the MENA region. However, the availability of systematic reviews that comprehensively collect and analyze this data is limited. Conducting a systematic literature review will help bridge this gap by providing a comprehensive synthesis of the existing research, allowing for a more thorough understanding of the epidemiology of SUDs and MHDs among this population.”

2. Inclusion/exclusion criteria:

a. The authors’ definition of “forced migrants displaced from the MENA region” could be sharpened. Taken at face value, this wording suggests that the authors will only include studies on people that have been displaced from the region entirely, which would exclude the steep majority, who have been displaced within the region. From the search terms, I don’t believe this is the authors’ intention, but it would be worth stating in the inclusion/exclusion criteria section exactly what they intend. For instance: studies focused on people from the MENA region who have been forcibly displaced by conflict, persecution, and/or natural disaster, including international displacement as well as internal displacement. 

Response to the reviewer’s comment: Thank you for your valuable feedback. We intended to include both internationally and internal displacement, so we included this sentence per your suggestion. Please see page 6, lines 140-142: “Studies focused on people from the MENA region who have been forcibly displaced by conflict, persecution, and/or natural disaster, including international displacement as well as internal displacement will be included.”.

Relatedly, the authors’ definition of the MENA region is a little fuzzy, since UNOCHA responds to both places producing displacement (e.g., Syria) and places receiving displaced people (e.g., Jordan). Often, for example, Turkey is not included in definitions of the MENA region (for instance, in OHCHR), but UNOCHA includes it because of its Syrian response.

Response to the reviewer’s comment: Thank you for your insightful comment. We used United Nations Human Resources Office of the High Commissioner (HROHC) classification for included countries in MENA region plus Türkiye (MENAT).

Some related questions: Is there a minimum/maximum length of time for the length of displacement that the authors will include? For instance, will they include resettled refugees, who are technically no longer displaced? Will they include people who fled a natural disaster for a month, only to return? Will they include returnees more generally?

Response to the reviewer’s comment: There is no specific length of time for displacement in this study. All refugees resettled or not resettled will be included since even after resettlement they might struggle with SUDs and/or MHDs, we decided to include them all.

b. Will the authors include studies that include participants from other regions, or only studies that focus exclusively on migrants from the MENA region?

Response to the reviewer’s comment: Only studies on forced migrants from the MENA region. Expanding the inclusion criteria to encompass forced migrants from other regions would significantly increase the number of extracted articles, making the process of data extraction and synthesis unmanageable. Furthermore, there already exists a substantial body of research focusing on MHDs among forced migrants specifically from the MENA region.

c. The authors justify the 2013-2023 period with a parenthetical reference to the Arab Spring, but it’s unclear how this fits exactly. The Conventional periodization of the Arab Spring is 2010-2012, so is the goal to include only studies that followed the Arab Spring? Why not also include 2010-2012? As an aside, the introduction section attributes displacement to the Arab Spring protest, rather than to the state crackdowns, coups, and war that followed. I would consider reframing.

Response to the reviewer’s comment: We aim to include the studies conducted in the last ten years not more to make the process of data extraction and synthesis manageable. The introduction section is reframed per your suggestion. Please see page 3, lines 59-62: “The Arab Spring protest and the subsequent state crackdowns, coups, and wars that followed the initial protests played a significant role in the displacement such that caused more than 3.5 million people to get internally displaced; ever since the numbers have even increased to more than three times.”

d. Will the authors include qualitative, quantitative studies, and mixed-methods studies, or only quantitative studies?

Response to the reviewer’s comment: All qualitative, quantitative studies, and mixed-methods studies will be included. We added this information to the methods section. Please see page 6, lines 133-134: “The systematic review will include all qualitative, quantitative studies, and mixed-methods studies focusing on SUDs and/or MHDs among forced migrants displaced from the MENA region.”

3. Search strategy:

a. Concept 1: Consider adding: Türkiye (official name), Kurdistan, and Near East

b. Concept 2: Depending on your responses to the above, consider adding asylee, resettled, and returnee

c. Concept 3: Consider adding regionally specific substances, such as khat/qat, hookah/narghile/argila/shisha/waterpiped. Consider adding search terms in Arabic

Response to the reviewer’s comment : Thank you for these suggestions about additional search terms. We have added them all, as well as demonyms for Turkish and Kurdish people, and some additional synonyms/spellings for the regionally specific substances.

Regarding the suggestion of adding search terms in Arabic: we considered this approach, but we decided not to implement it. We are confident that we can nevertheless expect to identify relevant Arabic-language papers with the current approach, through three pathways. First, many Arabic-language papers in the databases we are searching have English-language abstracts, through which our search strategy will retrieve them. Second, Arabic-language papers in the databases which we are searching with controlled vocabulary will be retrieved by our use of subject headings and our use of the Ovid multipurpose field (as opposed to the Ovid textword field). Third, papers (in any language) in journals that are not indexed in any of the databases which we searched can still be identified in this project through citation chaining, which we will conduct with the tool Citation Chaser (Haddaway 2022), using the bibliographic database Lens, which includes publication metadata from the database/dataset OpenAlex (https://about.lens.org/release-8-5/), which is recognized as having particularly good coverage of open access journals based in LMIC countries (Khanna 2022). Via these pathways, we hope to identify relevant papers in Arabic, even if they are from journals not indexed in any of the bibliographic databases we are searching. What’s more, we intend to create a supplemental table of relevant articles in other languages – Persian, French, Turkish, any other language – that we identify during the screening process, for use by other researchers studying this topic in the future. 

Khanna, S., Ball, J., Alperin, J. P., & Willinsky, J. (2022). Recalibrating the Scope of Scholarly Publishing: A Modest Step in a Vast Decolonization Process. Quantitative Science Studies, 1–43. https://doi.org/10.1162/qss_a_00228

Haddaway, N. R., Grainger, M. J., & Gray, C. T. (2022). Citationchaser: A tool for transparent and efficient forward and backward citation chasing in systematic searching. Research Synthesis Methods, n/a(n/a). https://doi.org/10.1002/jrsm.1563

Reviewer #2: Thank you for inviting me to review the protocol “Epidemiology of substance use and mental health disorders among forced migrants 2 displaced from the MENA region: a systematic review protocol”. Overall, I believe that the review topic itself has merit, but this manuscript would benefit from major revisions, and perhaps even the assistance of an editor for help with writing as there is awkward phrasing throughout the paper. I have provided my comments as both a topic expert (migrant health) and as a systematic review methodologist. I hope the authors find my comments useful in preparing their next version.

ABSTRACT

The abstract requires substantial revisions – I suggest that the authors review the PRISMA 2020 for Abstracts Checklist (http://prisma-statement.org/Extensions/Abstracts) to help with the overall reporting and structure. Importantly, the authors should provide an explicit statement of the main objective(s) or question(s) the review addresses in the background and define all acronyms at their first mention (e.g., “SU”, “MH”, “MENA”). The Methods should describe the inclusion criteria and the planned approach for synthesis.

Response to the reviewer’s comment: Thank you for your constructive feedback. The Abstract is revised and improved now.

INTRODUCTION

Major:

- Please review the UNHCR statistics and verify if more recent stats are available (2022? 2021?).

Response to the reviewer’s comment: Data is updated from UNHRC.

- Many statements are provided without in-text citations (e.g., Lines 51-52; 63-71). Please provide supporting references.

Response to the reviewer’s comment: References added.

Minor:

- Lines 77 and 78 are repetitive.

Response to the reviewer’s comment: Thank you for noticing that. This sentence is modified.

METHODS

Major:

Given the broad nature of the review questions presented by the authors, and the descriptive planned synthesis and presentation of the results, I strongly recommend the authors to consider whether a scoping review is more appropriate review choice. The authors should justify their choice in their protocol. There are resources available on making this decision, as well as guidance for conduct and reporting:

- Munn, Z., Peters, M. D., Stern, C., Tufanaru, C., McArthur, A., & Aromataris, E. (2018). Systematic review or scoping review? Guidance for authors when choosing between a systematic or scoping review approach. BMC medical research methodology, 18, 1-7. https://bmcmedresmethodol.biomedcentral.com/articles/10.1186/s12874-018-0611-x- JBI MANUAL FOR EVIDENCE SYNTHESIS: SCOPING REVIEWS CHAPTER: https://jbi.global/scoping-review-network/resources

- Tricco, AC, Lillie, E, Zarin, W, O'Brien, KK, Colquhoun, H, Levac, D, Moher, D, Peters, MD, Horsley, T, Weeks, L, Hempel, S et al. PRISMA extension for scoping reviews (PRISMA-ScR): checklist and explanation. Ann Intern Med. 2018,169(7):467-473. doi:10.7326/M18-0850. http://www.prisma-statement.org/Extensions/ScopingReviews

Response to the reviewer’s comment: We appreciate the reviewer's thoughtful consideration of the review questions and the recommended resources. We have thoroughly reviewed the suggestions and have reevaluated the appropriateness of the review type for each question.

After careful consideration, we agree that the primary question (PQ1) has a broad nature and requires a systematic review approach. We will conduct a comprehensive and methodical search of the literature to identify all relevant studies, followed by a rigorous synthesis and analysis of the findings. A systematic review will allow us to address PQ1 with a high level of evidence and ensure a robust and reliable conclusion.

Regarding the secondary questions, SQ1 and SQ2, and primary question 2 (PQ2) we acknowledge that they might benefit from a scoping review approach due to their exploratory and broad scope nature. A scoping review will enable us to map the available evidence, identify key concepts, and provide an overview of the literature without necessarily assessing the quality of individual studies. This approach will help us gain a better understanding of the existing research landscape related to these questions.

In our protocol, we will explicitly specify the review type for each question and provide a clear justification for our choice. We will also refer to the recommended resources, such as the Munn et al. (2018) paper on choosing between systematic and scoping reviews and the PRISMA-ScR checklist by Tricco et al. (2018), to ensure that our review follows best practices for conduct and reporting.

Please see page 14, lines 240-252: “We aim to investigate a comprehensive range of research questions to gain a thorough understanding of the topic under study. The primary question (PQ1) has been identified as having a broad nature, necessitating a systematic review approach. By following a systematic review methodology, we will conduct a rigorous and exhaustive search of the literature, ensuring the inclusion of all relevant studies and providing a robust synthesis and analysis of the findings.

Additionally, we recognize that PQ2, SQ1, and SQ2 might benefit from a scoping review approach due to their exploratory and broad scope nature. Employing a scoping review methodology for these secondary questions will enable us to map the available evidence, identify key concepts, and offer an overview of the literature without necessarily assessing the quality of individual studies. This approach will provide a comprehensive understanding of the existing research landscape related to these questions and facilitate the identification of knowledge gaps and potential areas for future research.”.

Under “Participants”, the authors should define the term “forced migrants”. For example, will the authors include migrants who cross a border (refugees, asylum seekers) and those who don’t (IDPs?). What types of migrant populations would be excluded (e.g., economic migrants, international students, other?).

Response to the reviewer’s comment: The definition added per your suggestion.

If the authors choose to conduct a systematic review, then they must specify interventions and outcomes eligible for inclusion.

Response to the reviewer’s comment: Thank you for your comment. This systematic review is a systematic review of prevalence and incidence. Including interventions and outcomes is not applicable; it is more applicable for Cochrane-style systematic review of interventions.

Under “Risk of Bias Assessment” – the authors have selected the Cochrane RoB 2.0 tool to assess intervention studies, however this tool it specifically designed for RCTs and does not adequately assess other non-randomized study designs. The authors should consider ROBINS-I. 

Response to the reviewer’s comment: Thank you for your suggestion. Edited!

Given that the studies addressing review question #1 are likely to be observational and reporting prevalence, the authors should also consider whether the JBI critical appraisal checklist for prevalence studies would be beneficial. Additional guidance for reporting such evidence is available:

- Munn Z, Moola S, Lisy K, Riitano D, Tufanaru C. Methodological guidance for systematic reviews of observational epidemiological studies reporting prevalence and incidence data. Int J Evid Based Healthc. 2015;13(3):147–153.

Response to the reviewer’s comment: Thank you for your helpful comment. We will use the JBI critical appraisal checklist for prevalence study that you mentioned. The manuscript’s content edited accordingly.

Under “Data synthesis” it’s not clear why the authors have not described a synthesis approach for the pooling of prevalence estimates, which would be expected in a systematic review addressing question #1. The charting approach described by the authors would be better suited to a scoping review.

Response to the reviewer’s comment: The approach to synthesizing prevalence studies is added to the manuscript per your suggestion: “The ‘best evidence synthesis’ will be used to synthesizing prevalence estimates. In implementing a best evidence synthesis approach, this study will prioritize higher-quality studies or those with more robust methodology to guide the synthesis process [29]. Greater weight will be assigned to studies with larger sample sizes, rigorous study designs, or lower risk of bias when summarizing prevalence estimates. By employing this approach, we aim to provide a more reliable and valid synthesis of prevalence data that will be informed by studies of the highest quality and methodological rigor.”

29- Slavin, R.E., Best-Evidence Synthesis: An Alternative to Meta-Analytic and Traditional Reviews. Educational Researcher, 1986. 15(9): p. 5-11.

The authors previously described that “studies with high risk of bias will be considered in a sensitivity analysis”, but the methods for this analysis are not described in this section.

Response to the reviewer’s comment: We appreciate the reviewer's comment. We added a description about methods for sensitivity analysis.

Given the explicit decision not to pool outcomes, the authors will find it challenging to conduct GRADE assessments, as there may be a large number of single-study estimates. Additionally, the authors have not explicitly defined any outcomes in their eligibility criteria or analysis plan, and so it is unclear what types of outcomes will be assessed using GRADE. Notably, there is currently no GRADE guidance available for the assessment of the certainty of evidence of prevalence data. The authors should describe how they plan to assess the evidence for review question 1.

Response to the reviewer’s comment: Thank you for your comment. We will use ‘best evidence synthesis’ to synthesizing prevalence estimates as mentioned above.

Minor:

- Line 97 – it is not necessary to describe PROSPERO and cite Booth et al., your reference should include the citation information necessary to locate your registration.

Response to the reviewer’s comment: We have included registration ID under the “Design of the systematic review” section. Citation for PROSPERO removed.

- There is a word missing, Line 99: “The current ___ uses the framework…”

Response to the reviewer’s comment: Edited! Thank you.

- The headings “study selection criteria” and “inclusion and exclusion criteria” are redundant – The PRISMSA-P Checklist uses the language “Eligibility criteria” – the authors should review PRISMA-P and ensure that they are adequately reporting all items.

Response to the reviewer’s comment: Edited! Thank you.

- Under “screening”, the authors should specify that screening will be done in duplicate

Response to the reviewer’s comment: Added.

---

## [Editor Report · Decision Letter 2]

16 Aug 2023

PONE-D-23-12703R2Epidemiology of substance use and mental health disorders among forced migrants displaced from the MENAT region: a systematic review protocolPLOS ONE

Dear Dr. Kazemitabar,

Thank you for submitting your manuscript to PLOS ONE. After careful consideration, we feel that it has merit but does not fully meet PLOS ONE’s publication criteria as it currently stands. Therefore, we invite you to submit a revised version of the manuscript that addresses the points raised during the review process.

ACADEMIC EDITOR: The response letter remains unchanged from the previous version of the manuscript, and the new comments from the reviewers have not yet been addressed in the revised manuscript (R2 version). Please thoroughly address the reviewers' comments and submit the revised version along with the response letter for further consideration.

We look forward to receiving your revised manuscript.

Kind regards,

Amin Nakhostin-Ansari

Academic Editor

PLOS ONE

---

## [Author Response · Author response to Decision Letter 2]

5 Sep 2023

September 5, 2023

Dear Reviewers,

Thank you for your insightful comments and feedback on our manuscript which resulted in improving it. We addressed your comments and incorporated them in this revised version. Please see below our responses to your comments.

Sincerely,

Maryam Kazemitabar

Please see our responses to the reviewers’ comments below. My responses to the reviewers' comments are in bold black and the contents in bold blue are those that directly refer to the manuscript.

Reviewer #1: The authors have thoroughly addressed my comments and produced an improved submission. I have two small remaining notes: First, I find the justification for the systematic review for PQ1 somewhat confusing. In response to Reviewer 2, the authors note that "After careful consideration, we agree that the primary question (PQ1) has a broad nature and requires a systematic review approach." They then write in the body of the text that "The primary question (PQ1) has been identified as having a broad nature, necessitating a systematic review approach." This is further confused by the fact that the authors also justify a *scoping* review for PQ2, SQ1, and SQ2 on the basis of their "broad scope nature." I disagree that "having a broad nature" is what warrants a systematic review, for PQ1, and I believe Reviewer 2 actually suggested a *scoping* review (rather than a systematic review) specifically "given the broad nature of the review questions." This is in keeping with the Munn et al. article on deciding between a systematic and scoping review, where they note that "A key difference between scoping reviews and systematic reviews is that in terms of a review question, a scoping review will have a broader “scope” than traditional systematic reviews..." That said, I agree with the authors that a systematic review is appropriate for PQ1—given the question's focus on synthesizing evidence of prevalence—but I would suggest articulating the justification more clearly. For example, using the same Munn et al. article, the authors could argue that PQ1 meets the systematic review indication for "uncover[ing] the international evidence." My second small comment is that some of the edits have introduced new grammatical errors (e.g. page 3, line 75), so I would recommend a comprehensive copy edit before publishing.

Response to the reviewer’s comment: Thank you for your helpful comment. It is edited according to your suggestion. Please see page 15 line 276. We also reviewed and edited the whole manuscript for English proficiency.

Reviewer #2: Thank you for addressing my previous comments and for inviting me to review the revised paper. I appreciate the efforts the authors have made in their protocol. Below, I have a few additional comments that should be addressed prior to publication.

1) The abstract states that you will use the Newcastle-Ottawa Scale and Cochrane RoB, but the methods have been revised to the JBI checklist for prevalence studies and ROBINS-I. This needs to be consistent.

Response to the reviewer’s comment: Good catch! Thank you for reminding us to edit this in the Abstract.

2) More details are needed regarding how you plan to analyze the data for PQ1. For example, do you plan to conduct a meta-analysis of prevalence estimates? If yes, will you use a random or fixed effects model? Will these estimates be stratified by MH condition? Will you conduct any assessments for publication bias? You do describe the best evidence synthesis approach to weight studies in your analyses, but it is still necessary to describe the analytic approach. I would consider reviewing the paper by Blackmore et al. below as an example of the type of analyses that could be possible. If the authors choose not to conduct a meta-analysis (or other regression analyses), this choice needs to be justified in the protocol.

Citation: Blackmore R, Boyle JA, Fazel M, Ranasinha S, Gray KM, Fitzgerald G, et al. (2020) The prevalence of mental illness in refugees and asylum seekers: A systematic review and meta-analysis. PLoS Med 17(9): e1003337. https://doi.org/10.1371/journal.pmed.1003337

Response to the reviewer’s comment: Thank you for your valuable suggestion. We considered a meta-analysis for PQ1, focusing on SUDs and MHDs prevalence per your suggestion. Please see pages 14-15, lines 249-271:

“Statistical analysis

The meta-analysis model will be chosen based on the study design. Given the expected variability among studies, a random-effects model will be considered due to its capacity to account for heterogeneity. If utilizing the ‘meta’ package, the ‘metagen’ function [1] will be applied. Alternatively, for the ‘metafor’ package, the ‘rma’ function [2] will be used. The selected function will be executed by inputting a structured data frame and defining relevant parameters including effect size, standard error, sample size, and study identifiers.

To quantify heterogeneity among the studies, I² statistic [3] will be employed to quantify the proportion of total variation in effect estimates that is due to heterogeneity rather than chance. Visual representations of study effect sizes and confidence intervals will be generated through the creation of forest plots, using functions like ‘forest()’ or ‘forest.rma()’. Potential publication bias will be explored and visualized using funnel plots and Egger's test [4] considering a p-value < .05. These analyses will be facilitated through dedicated functions available in the chosen R package. 

Subgroup analyses will be conducted to find potential sources of heterogeneity by categorizing studies based on specific study characteristics. Variables anticipated to introduce heterogeneity will be considered for these analyses. Finally, for each variable, separate subgroups will be created, grouping studies with similar characteristics together. The meta-analysis model will then be applied within each subgroup to calculate the pooled effect estimate and associated confidence intervals. Comparing the effect estimates across different subgroups will allow for the identification of patterns or trends, revealing how each variable may impact the overall results. This process will provide valuable insights into the relative contribution of each characteristic to the observed heterogeneity.”

We also updated the abstract to reflect these changes.

3) The authors have still not addressed the issue with planning to use GRADE methods. Given that the outcomes for PQ2 and SQ1/2 will be scoping review questions and no risk of bias assessments will be complete, no GRADE assessment will be possible for these questions. Additionally, there are no current methods for GRADE assessments for prevalence data (PQ1). My suggestion would be for the authors to acknowledge these limitations and state that no assessment for the overall certainty/strength of the evidence will be performed.

Response to the reviewer’s comment: Thank you for your feedback. We appreciate your input and have taken it into consideration. As per your suggestion, we have included the following sentence to address the issues you mentioned (page 14, lines 251-253):

“Given that the outcomes for PQ2/3 and SQ1/2 will be scoping review questions and no risk of bias assessments will be completed, there will be no evaluation conducted to determine the overall certainty or strength of the evidence.”

4) In response to reviewers #1's comments, the authors described "papers (in any language) in journals that are not indexed in any of the databases which we searched can still be identified in this project through citation chaining, which we will conduct with the tool Citation Chaser (Haddaway 2022), using the bibliographic database Lens, which includes publication metadata from the database/dataset OpenAlex (https://about.lens.org/release-8-5/), which is recognized as having particularly good coverage of open access journals based in LMIC countries (Khanna 2022)."

These details (such as your planned use of the tool citation chaser) should be described in the protocol and referenced appropriately. You should list the software you plan to use for your analysis (e.g., RevMan, STATA, R) and any planned visualizations.

Response to the reviewer’s comment: Thank you for your insightful comments. We appreciate your attention to detail. We added the following information to the protocol per your suggestion:

“Additionally, research articles published in journals not covered by the databases we searched can still be located using citation chaining in this study. We will perform this process using the Citation Chaser tool [5] and the bibliographic database Lens. This database incorporates publication details from the OpenAlex database/dataset, known for its strong coverage of open-access journals in low- and middle-income countries [6]. Citation Chaser is an automated tool that streamlines the process of "citation chasing" in systematic reviews. It uses the Lens.org API to quickly retrieve lists of references from various studies and identify articles that cite a specific study. This eliminates the manual effort traditionally required for cross-referencing and enhances accuracy. The tool can generate lists of both referenced and citing records from sources like PubMed, PubMed Central, CrossRef, Microsoft Academic Graph, and CORE, making systematic review searches more efficient.”

27- Haddaway NR, Grainger MJ, Gray CT. Citationchaser: A tool for transparent and efficient forward and backward citation chasing in systematic searching. Research Synthesis Methods. 2022 Jul;13(4):533-45.

28- Khanna S, Ball J, Alperin JP, Willinsky J. Recalibrating the scope of scholarly publishing: A modest step in a vast decolonization process. Quantitative Science Studies. 2022 Dec 22:1-9.

The statistical analysis and the software and packages used to conduct meta-analysis described in details as explained in response to your previous comment.

References

1. Viechtbauer, W., Conducting meta-analyses in R with the metafor package. Journal of statistical software, 2010. 36: p. 1-48.

2. Viechtbauer, W. The R package metafor: Past, present, and future. in Research Synthesis 2019 incl. Pre-Conference Symposium Big Data in Psychology, Dubrovnik, Croatia. 2019. ZPID (Leibniz Institute for Psychology Information).

3. Higgins, J.P., et al., Measuring inconsistency in meta-analyses. Bmj, 2003. 327(7414): p. 557-560.

4. Egger, M., et al., Bias in meta-analysis detected by a simple, graphical test. Bmj, 1997. 315(7109): p. 629-634.

5. Haddaway, N.R., M.J. Grainger, and C.T. Gray, Citationchaser: A tool for transparent and efficient forward and backward citation chasing in systematic searching. Research Synthesis Methods, 2022. 13(4): p. 533-545.

6. Khanna, S., et al., Recalibrating the scope of scholarly publishing: A modest step in a vast decolonization process. Quantitative Science Studies, 2022: p. 1-19.

---

## [Decision Letter · Decision Letter 3]

25 Sep 2023

Epidemiology of substance use and mental health disorders among forced migrants displaced from the MENAT region: a systematic review and meta-analysis protocol

PONE-D-23-12703R3

Dear Dr. Kazemitabar,

We’re pleased to inform you that your manuscript has been judged scientifically suitable for publication and will be formally accepted for publication once it meets all outstanding technical requirements.

Kind regards,

Amin Nakhostin-Ansari

Academic Editor

PLOS ONE

Additional Editor Comments (optional):

Reviewers' comments:

Reviewer's Responses to Questions

**Comments to the Author**

1. Does the manuscript provide a valid rationale for the proposed study, with clearly identified and justified research questions?

Reviewer #1: Yes

2. Is the protocol technically sound and planned in a manner that will lead to a meaningful outcome and allow testing the stated hypotheses?

Reviewer #1: Yes

3. Is the methodology feasible and described in sufficient detail to allow the work to be replicable?

Reviewer #1: Yes

4. Have the authors described where all data underlying the findings will be made available when the study is complete?

Reviewer #1: No

5. Is the manuscript presented in an intelligible fashion and written in standard English?

Reviewer #1: Yes

6. Review Comments to the Author

You may also provide optional suggestions and comments to authors that they might find helpful in planning their study.

Reviewer #1: The authors have done a great job reviewing the manuscript and I wish them all the best with their study.

7. PLOS authors have the option to publish the peer review history of their article (what does this mean?). If published, this will include your full peer review and any attached files.

Reviewer #1: **Yes: **Cyril Bennouna

---

## [Editor Report · Acceptance letter]

27 Sep 2023

PONE-D-23-12703R3 

Epidemiology of substance use and mental health disorders among forced migrants displaced from the MENAT region: a systematic review and meta-analysis protocol 

Dear Dr. Kazemitabar:

I'm pleased to inform you that your manuscript has been deemed suitable for publication in PLOS ONE. Congratulations! Your manuscript is now with our production department. 

Kind regards, 

on behalf of

Dr. Amin Nakhostin-Ansari 

Academic Editor

PLOS ONE